# The Duration of Emotional Image Exposure Does Not Impact Anticipatory Postural Adjustments during Gait Initiation

**DOI:** 10.3390/brainsci8110195

**Published:** 2018-11-01

**Authors:** Thierry Gélat, Laure Coudrat, Carole Ferrel-Chapus, Sylvie Vernazza-Martin

**Affiliations:** 1Centre de Recherche sur le Rport et le Mouvement, EA 2931, UFR STAPS Université Paris Ouest Nanterre, 200 Avenue de la République, 92001 Nanterre, CEDEX, France; cferrelc@u-paris10.fr (C.F.-C.); sylvie.martin@u-paris10.fr (S.V.-M.); 2Laboratoire Lorrain de Psychologie et Neurosciences de la Dynamique des Comportements, 2LPN, CEMA Group (Cognition-EMotion-Action), EA 7489, Université de Lorraine, 57000 Metz, France; laure.coudrat@wanadoo.fr

**Keywords:** emotion, gait initiation, anticipatory postural adjustments, duration of images exposure

## Abstract

Previous studies have reported that anticipatory postural adjustments (APAs) associated with gait initiation are affected by emotion-eliciting images. This study examined the effect of the duration of exposure to emotional images on the APAs along the progression axis. From a standing posture, 39 young adults had to reach a table by walking (several steps) toward pleasant or unpleasant images, under two sets of conditions. In the short condition, the word “go” appeared on the image 500 ms after image onset and participants were instructed to initiate gait as soon as possible after the word go appeared. In the long condition, the same procedure was used but the word “go” appeared 3000 ms after image onset. Results demonstrated that the APAs were longer and larger for pleasant images than unpleasant ones, regardless of the condition (i.e., the duration of exposure to the images). In the same way, the peak of forward velocity of the centre of body mass (reached at the end of the first step) followed the same tendency. These results emphasized that APAs depended on image valence but not on the duration of images exposure and were consistent with those of previous studies and the motivational direction hypothesis.

## 1. Introduction

It is well known that a voluntary movement needs to be coordinated with posture in order to be performed efficiently [1]. Indeed, the efficiency of the postural phase associated with an intended movement may be assessed via the spatial and temporal characteristics of anticipatory postural adjustments (APAs). From a general point of view, the latter are crucial for the control of balance during daily motor tasks to counteract the destabilizing effect induced by a predicted disturbance [2].

During gait initiation, the anticipatory phase begins with a postural movement and its role is to destabilize the whole body prior to the execution phase, beginning at heel-off by the stepping limb and ending at foot contact of this limb. Destabilization of the erect posture by APAs occurs by displacing the centre of pressure (CP) backwards and onto the stepping limb, enabling the centre of body mass (CM) to be propelled forwards and onto the stance limb. APAs are organized subcortically and the systems responsible for their execution are separate from those that control the goal-directed voluntary movement [3]. In gait initiation, they are modulated according to the forthcoming requirements of the task, for example, the gait velocity [4,5] or the balance requirements in the frontal plane [6].

APAs are also known to depend on the emotional context during gait initiation. Emotion is considered to prime the body for action [7]. According to the biphasic theory of emotion, emotion is fundamentally organized around two basic motivational systems: the appetitive and defensive system, priming approach and avoidance behaviours respectively [8]. Using emotional images, it was reported that when participants had to initiate forward gait as soon as possible, at image onset [9] or offset, that is, when the image presented for a variable 2–4 s duration, disappeared [10], APAs on the progression axis were greater with pleasant than with unpleasant images. These results supported the motivational direction hypothesis [8], emphasizing that approach movement towards a pleasant stimulus was facilitated, whereas approach movement towards an unpleasant stimulus was impeded. We will call a valence effect the result that APAs were larger with pleasant images, as compared to unpleasant images.

Event-related potential studies provided evidence that the valence of an emotional image is processed very quickly (100–250 ms), whereas arousal is processed later (250–850 ms) [11]. However, one may wonder whether duration of emotional image exposure for longer than this range interacts with gait parameters. To date, little is known on how and why the effect of image valence evolves over time. For instance, we know during gait initiation that reaction time is shorter with pleasant than unpleasant images at image onset [9,12] or when the image exposure duration was short (0.5 s) [13]. In contrast, the reverse was found at image offset (the image was presented for 2–4 s) [10]) or there was no difference between both image categories when the image exposure duration was long (3 s) [13].

Comparing an image onset situation to an offset one during gait initiation allowed the examination of the impact of the duration of emotional image exposure on APAs. Using a single step, it was reported that the backward shift of CP (APAs amplitude) during the anticipatory phase was found to be larger at image offset (the image was presented for a variable 2–5 s duration) than image onset [14]. Recently, it was reported that when the delay between image onset and the imperative “go” was either 0.5 s (short condition) or 3 s (long condition), the initial posture preceding the movement was modified in the long condition as compared to the short one [13]. Indeed, it was found in this study that the whole body leaned forwards and onto the stance limb prior to initiating the movement during the long (but not the short) condition, regardless of the valence of the image. Prior to gait initiation, a forward leaning of the body usually aims to increase the peak of CM velocity, as already reported [4,15]. Hence, we reasoned that APAs on the progression axis might be larger from the short to the long condition because the body leaned forwards when the delay between image onset and the imperative “go” was long.

The main aim of this study was to examine whether the APAs on the anteroposterior axis during gait initiation were affected by the duration of emotional image exposure prior to the imperative “go”. As in a previous study [13], we used a reaction time paradigm in which the delay between image onset and the imperative “go” was either 0.5 s in the short condition or 3 s in the long condition. Reaction time (RT) was measured to know whether results were in line with those found in a previous study [13]. We expected RT to be shorter in the long than in the short condition. Moreover, RT would be shorter with pleasant than with unpleasant images in the short condition but that no significant difference between both emotional categories would be found in the long condition, as already reported [13]. Concerning the APAs, one hypothesis was that they might be larger from the short to the long condition (i.e., a condition effect), because a previous study demonstrated that the body leaned forwards prior to the movement in the long condition. Alternatively, only a valence effect might be expected for APAs. As subcortical regions are responsible for the control of APAs in gait initiation, an automatic process may be assumed to be involved in this control. Moreover, APAs have been reported to be progressively constructed early before the onset of the intended movement [16]. Hence, they might be independent of the time preceding the imperative “go” and a condition effect would not be expected. Furthermore, as emotional stimuli have been reported to be processed automatically [17], the amplitude of APAs would be expected to be greater with pleasant than with unpleasant images, according to the motivational direction hypothesis.

## 2. Methods

### 2.1. Participants

Thirty-nine young healthy adults (age 18–26 years, 22 women) recruited from the University of Paris Nanterre participated in this study. All were right handed, had normal or corrected-to-normal vision, were unaware of the purpose of the study and did not report any psychiatric disorders. The study conformed to the standards set by the Declaration of Helsinki and informed consent was obtained from all participants.

### 2.2. Experimental Protocol

We used a force platform (AMTI, 120 × 60 cm) that measured the three components of ground reaction forces (Fx, Fy and Fz) and moments (Mx, My and Mz). Participants stood on a force platform placed at the beginning of a 5 m long walkway ending at a white wall of the laboratory on which the emotion eliciting images (86 × 125 cm) were presented. The positioning of the feet allowed the first step to be performed on the platform and was marked to be used in all trials. On the medio-lateral axis, participants were instructed to place their feet on both sides of a line separating the platform into two equal parts. The distance between the feet had to not exceed the pelvis width. Participants had to walk toward an emotional image up to a table and click on the mouse situated on the table. It is noteworthy that no instruction was given to participants regarding gait velocity.

Two experimental conditions were tested. Each trial began with the presentation of a fixation cross on the screen for 2 s, followed by an image for 6 s. In the short condition, the word “go” appeared at the centre of the picture 0.5 s after image onset and participants were instructed to (1) look at the image for the entire time it was on the wall; (2) initiate gait with the right limb as soon as possible after the word go appeared and (3) reach the table to click on the mouse. This action triggered a computerized 9-point version of the self-assessment manikin (SAM) [18] procedure enabling to obtain subjective ratings of valence and arousal of pleasant and unpleasant images of this condition. In the long condition, the same procedure was used but the word “go” appeared 3 s after image onset.

### 2.3. Emotional Images

In each condition, participants viewed twenty images (10 pleasant and 10 unpleasant) selected from the international affective picture system (IAPS) [18]. As in a previous study [8], half of the pleasant and unpleasant images were low arousal (LA) and the other half were high arousal (HA), representing four affective categories: happy people for low arousal pleasant images, erotic for high arousal pleasant images, sad people for low arousal unpleasant images and attack for high arousal unpleasant images. The normative valence of pleasant and unpleasant images differed significantly (M = 7.23, SD = 0.49 for pleasant images; M = 2.38, SD = 0.40 for unpleasant images). The normative arousal of both image categories was similar, depending only on its level (low/high) (M = 4.98, SD = 0.53 for LA pleasant images; M = 6.35, SD = 0.26 for HA pleasant images; M = 5.09, SD = 0.41 for LA unpleasant images; M = 6.83, SD = 0.34 for HA unpleasant images). A statistical analysis on the valence and arousal of the pictures in each condition was performed to ensure that no significant difference was found between both conditions. The task was performed using a go/no-go paradigm. The no-go response (i.e., do not move) was associated with the presentation of 14 neutral images, representing an object. Each participant viewed 54 images that were randomly presented according to both the valence (pleasant, neutral, unpleasant) of images and the condition (short, long). E-prime 2.0 software (Psychology Software Tools, Pittsburg, PA, USA) was used to control the visual stimuli presentation of each trial.

### 2.4. Data Reduction

All signals from the force platform were recorded at 1000 Hz for 8 s and synchronized with the images presentation procedure. These signals were filtered with a 10 Hz low-pass, fourth order, zero-lag Butterworth filter. Coordinates of CP on the medio-lateral (ML) (x_P_) and anterior-posterior (AP) (y_P_) directions were calculated from the ground reaction forces and the moments measured by the force platform. The displacement of CP on AP direction was derived to obtain the CP velocity (y′_P_). It was noteworthy that 35 trials (4.6%) were rejected because either there was a considerable CP movement in the 500 ms window prior to image onset or, the t_0y_ value, corresponding to the reaction time (RT) defined as the delay between the onset of the imperative go (word “go”) and t_0y_ (Figure 1), was less than 200 ms or longer than 2000 ms. The body’s centre of mass (CM) acceleration on AP axis (y″_G_) was obtained from Newton’s law and the CM velocity (y′_G_) axis was calculated by simple integration, by posing that y″_G_ and y′_G_ equalled zero at t_0y_. That enabled the calculation of the peak of forward CM velocity, reached at the end of the first step. The times of toe off (TO_1_) and foot contact (FC) of the stepping limb and that of toe off of the stance limb (TO_2_) were obtained from the traces of x_P_ and y_P_ (Figure 1).

The initial posture (prior to image onset and the imperative “go”) was characterized by the CP position in the AP direction. It corresponded to the mean CP position during the 500 ms period preceding image onset (initial phase) and during the 500 ms period preceding the imperative go (final phase). The values were expressed as a percentage of base of support length. 0% corresponded to a CP position under the heels and 100% to a CP position under the toes.

### 2.5. Variables Analysed

The onset of postural modifications along the AP (t_0y_) axis was calculated as the time at which 10% of the first peak of y′_P_ was reached (Figure 1A), as in a previous study [9]. The initial posture was analysed by measuring the CP position during the initial and final phases of both conditions. During the initial posture, the participant is in equilibrium. Hence, the CP position may be assimilated into the vertical projection of the CM onto the ground. Reaction time (RT) was measured. The APAs duration along the AP axis corresponded to the time between t_0y_ and that of the maximal backward shift of CP (t_yPmin_) and the APA amplitude corresponded to the initial maximal backward shift of CP (y_Pmin_) (Figure 1B). The peak of forward CM velocity (V) and the time when it occurred (t_V_) were also analysed (Figure 1C). The length of the first step was calculated as the AP displacement of CP between t_0y_ and the time of toe off of the stance foot (TO_2_) (Figure 1B). Lastly, the movement time was calculated as the time between t_0y_ and the time when the participant clicked on the mouse situated on the table.

### 2.6. Statistical Analysis

The initial posture was analysed using a 2 (Condition: short, long) × 2 (Valence: pleasant, unpleasant) × 2 (Arousal: high, low) × 2 (Phase: initial, final) analysis of variance with repeated measures on the four factors. We used a 2 (Condition: short, long) × 2 (Valence: pleasant, unpleasant) × 2 (Arousal: high, low) analysis of variance with repeated measures on the three factors to analyse the other dependent variables of this study. Post hoc analyses were conducted with the HSD Tukey test. For all analyses, the alpha level for statistical difference was set at 0.05.

## 3. Results

### 3.1. SAM

Ratings of self-reported valence of pictures showed a main effect of condition, F (1,38) = 4.32, *p* < 0.05, η_P_^2^ = 0.102, indicating that pictures were rated as less unpleasant in the long (4.83 ± 2.55) than in the short (4.69 ± 2.70) condition. As expected, we found a valence effect F (1,38) = 511.28, *p* < 0.001, η_P_^2^ = 0.930, indicating that pleasant (7.13 ± 1.17) images were rated as more pleasant than unpleasant (2.39 ± 1.07) images. Importantly, we found an interaction between condition and valence F (1,38) = 14.10, *p* < 0.001, η_P_^2^ = 0.270. The post hoc analysis showed that the valence of unpleasant images was significantly (*p* < 0.001) higher in the long (2.55 ± 1.15) than in the short (2.23 ± 0.96) condition, whereas no significant (*p* = 0.886) difference was found with the valence of pleasant images between both conditions. The valence of pleasant images was 7.15 ± 1.22 and 7.10 ± 1.13 in the short and long condition, respectively. Also, there was an interaction between valence and arousal F (1,38) = 14.97, *p* < 0.001, η_P_^2^ = 0.282. The post hoc analysis indicated that the valence of low arousing pleasant images (7.44 ± 0.90) was significantly (*p* < 0.05) higher than arousal of high arousing pleasant images (6.81 ± 1.33). In contrast, no significant (*p* = 0.156) difference was found between valence of low arousing (2.19 ± 0.81) and high (2.60 ± 1.25) arousing unpleasant images.

Ratings of self-reported arousal of pictures showed a main effect of arousal, F (1,38) = 8.72, *p* < 0.01, η_P_^2^ = 0.186, indicating that arousal of high arousing pictures (5.67 ± 1.91) was higher than that of low arousing pictures (5.20 ± 1.96). Also, there was an interaction between valence and arousal F (1,38) = 21.99, *p* < 0.001, η_P_^2^ = 0.366. The post hoc analysis showed that arousal of high arousing pleasant pictures (6.17 ± 1.60) was significantly (*p* < 0.001) higher than that of low arousing pleasant pictures (5.05 ± 1.82). In contrast, no significant (*p* = 0.797) difference was found between arousal of low arousing (5.35 ± 2.08) and high (5.17 ± 2.06) arousing unpleasant images.

### 3.2. Reaction Time

As in a previous study [12], we found a significant main effect for Condition (F (1,38) = 21.05, *p* < 0.001, η_P_^2^ = 0.356); the participants were faster in the long than in the short condition. A significant main effect for Valence was also present (F (1,38) = 8.29, *p* < 0.01, η_P_^2^ = 0.179), indicating that participants were faster for pleasant than for unpleasant pictures. Importantly, a significant interaction between Condition and Valence (F (1,38) = 12.09, *p* < 0.01, η_P_^2^ = 0.241) was found. Post hoc analysis revealed that motor responses were significantly (*p* < 0.001) faster for pleasant (0.442 ± 0.083) than for unpleasant pictures (0.478 ± 0.104) in the short condition but no significant (*p* = 0.998) difference was found in the long condition (0.402 ± 0.076 and 0.401 ± 0.079 s for pleasant and unpleasant pictures respectively).

### 3.3. Initial Posture

Regarding balance conditions prior to the imperative “go” (during the initial phase) on the AP axis, a significant main effect for Condition (F (1,38) = 23.81, *p* < 0.001, η_P_^2^ = 0.385) indicated that the CP position was further ahead in the long (43.38 ± 7.62%) than in the short (42.11 ± 7.41%) condition. There was also a main effect of Phase (F (1,38) = 32.52, *p* < 0.001, η_P_^2^ = 0.461) indicating that the CP was further ahead during the final than during the initial phase. Importantly, an interaction between Condition and Phase (F (1,38) = 37.60, *p* < 0.001, η_P_^2^ = 0.497) showed that between the initial (42.12 ± 7.45%) and final (42.09 ± 7.40%) phases no change was observed in the short condition (*p* = 0.999), whereas the body was tilted forwards (*p* < 0.001) in the long condition. The values were 42.25 ± 7.61% and 44.51 ± 7.49% in the initial and final phase respectively. There was no effect of Valence.

### 3.4. APAs

The APA duration on the progression axis only showed a valence effect (F (1,38) = 7.54, *p* < 0.01, η_P_^2^ = 0.165) indicating that it was longer for pleasant than for unpleasant images (see t_yPmin_ in Table 1). Similarly, the APA amplitude only showed a valence effect (F (1,38) = 35.39, *p* < 0.001, η_P_^2^ = 0.482) for which it was larger for pleasant than for unpleasant images (see y_Pmin_ in Table 1).

### 3.5. CM Velocity

The peak of CM forward velocity only indicated a main effect of valence (F (1,38) = 17.03, *p* < 0.001, η_P_^2^ = 0.309), showing that it was larger for pleasant than for unpleasant images (Table 1). No effect was found in the time to reach the peak of CM forward velocity (t_V_) (Table 1).

### 3.6. First Step Length

The length of the first step showed a valence effect (F (1,38) = 6.00, *p* < 0.05, η_P_^2^ = 0.136) indicating that it was larger for pleasant images as compared to unpleasant ones (Table 1).

### 3.7. Movement Time

A valence effect (F (1,38) = 15.94, *p* < 0.001, η_P_^2^ = 0.295) was found in the movement time showing that it was shorter for pleasant than for unpleasant images (Table 1).

## 4. Discussion

Using an RT paradigm with two conditions for which the delay between image onset and the imperative “go” was either short (0.5 s) or long (3 s), the aim of this work was to examine whether APAs on the AP axis associated with gait initiation were modified depending on the duration of emotional image exposure prior to the imperative “go”.

In line with our second hypothesis, only a valence effect was found in APAs. Indeed, results showed that the APAs duration was longer and the APAs amplitude was larger with pleasant as compared to unpleasant images, regardless of the condition. Consistent with this finding, a valence effect was also found in the peak of forward CM velocity, larger with pleasant as compared to unpleasant images. Moreover, the longer length of the first step and the shorter movement time (time to reach the table) observed with pleasant than with unpleasant images also showed only a valence effect.

All these findings clearly showed that no condition effect was present in the APAs and thus, participants did not take advantage of the change in the initial posture in the long condition, that is, a leaning forwards of the whole body. Yet, several studies reported that healthy participants modified their initial posture depending on the intended gait velocity [4,15]. In the latter study, young adults were instructed to initiate gait at three velocities: slow, normal and fast. Results showed that the participant’s forward leaning was greater in the fast condition than in the other ones. In the current study, the mean CP position (expressed in percentage of the support base length) during the 500 ms period preceding the imperative “go” in the long condition was about 45%. In the study by Lepers and Brenière [15], the initial forward leaning of the whole body corresponded to a CP position reaching 50% of the foot length, prior to the intended fast gait initiation. So, while the forward leaning of the body in both studies was very close, the following question arises: why did the advantage of a forward leaning of the body in the long condition not allow participants to make the APAs amplitude and thus the peak of CM forward velocity greater in the long than the short condition? We suggest that the following reasons may explain why no change in the APAs was found between the two conditions in this study. First, the movement goal between these two studies was quite different. Indeed, participants of the current study had to reach the table with no instruction regarding the gait velocity, whereas the aim of the movement in the study by Lepers and Brenière [15] was to initiate a fast gait. Then, we reasoned that the forward leaning of the initial posture during the long condition of this study might be viewed as a preparation of the body for action [7], reflecting an increased readiness of the participants for an approach behaviour but not the intention of a fast gait. Importantly, it has been reported from a ballistic wrist extension that when a high uncertainty in the timing of the imperative go was present, participants planned the movement well in advance [19]. Hence, we reasoned in this study that when the image appeared, the high uncertainty in the timing (short or long) of the imperative “go” with respect to the image onset would entail participants planning the movement well in advance too. As the APAs have been suggested to be implemented automatically as emotional images are processed [17], we reasoned that the APAs amplitude was determined depending only on the image valence. Hence, larger APAs were constructed with pleasant as compared to unpleasant images according to the motivational direction hypothesis. Furthermore, no effect of image arousal or interaction with condition or valence was found on APAs. This result is in contrast to that of an event-related potential study asking the participants to respond with a mouse-click when an emotional image was presented and for which arousal effects but attenuated valence effects, were found [20]. That likely meant that the current task (clear approach behaviour) needed a judgment of the valence of the affective image content. Hence, in line with the results of a previous onset study [9] and those of an offset one [10], APAs on the progression axis of the current study depended on the image valence.

## 5. Conclusions

The main result of this study was that during gait initiation APAs and the peak of forward velocity reached at the end of the first step did not depend on the duration of affective image exposure but only on the valence of these images. Moreover, no effect of the arousal level of these images was found, suggesting that the demands of the current task (approach to a pleasant/unpleasant image) especially required a judgment of the image valence [20]. Interestingly, APAs results of the current study markedly contrasted with those of reaction time. Indeed, the latter showed a condition effect and an interaction between condition and valence. They were found to be shorter in the long, as compared to the short condition and were similar with both emotional categories only in the long condition, as previously presented [13]. The later suggested that the onset of APAs was similar with pleasant as compared to unpleasant images in the long condition, because of the implementation of an implicit emotion regulation with unpleasant images in this condition, as previously reported [13]. An additional reason in favour of the implementation of an emotion regulation with unpleasant images in the long condition was that unpleasant images were judged as less unpleasant in the long that in the short condition while the judgment of pleasant images did not change across both conditions. Future studies will have to better examine this cognitive strategy. The fact that the reaction time was similar with pleasant and unpleasant images in the long condition may be viewed as an advantage of this condition as compared to the short condition. However, it was lost because the APAs amplitude was determined only by the image valence, making the movement faster with pleasant than unpleasant images, regardless of the duration of affective images.

## Figures and Tables

**Figure 1 brainsci-08-00195-f001:**
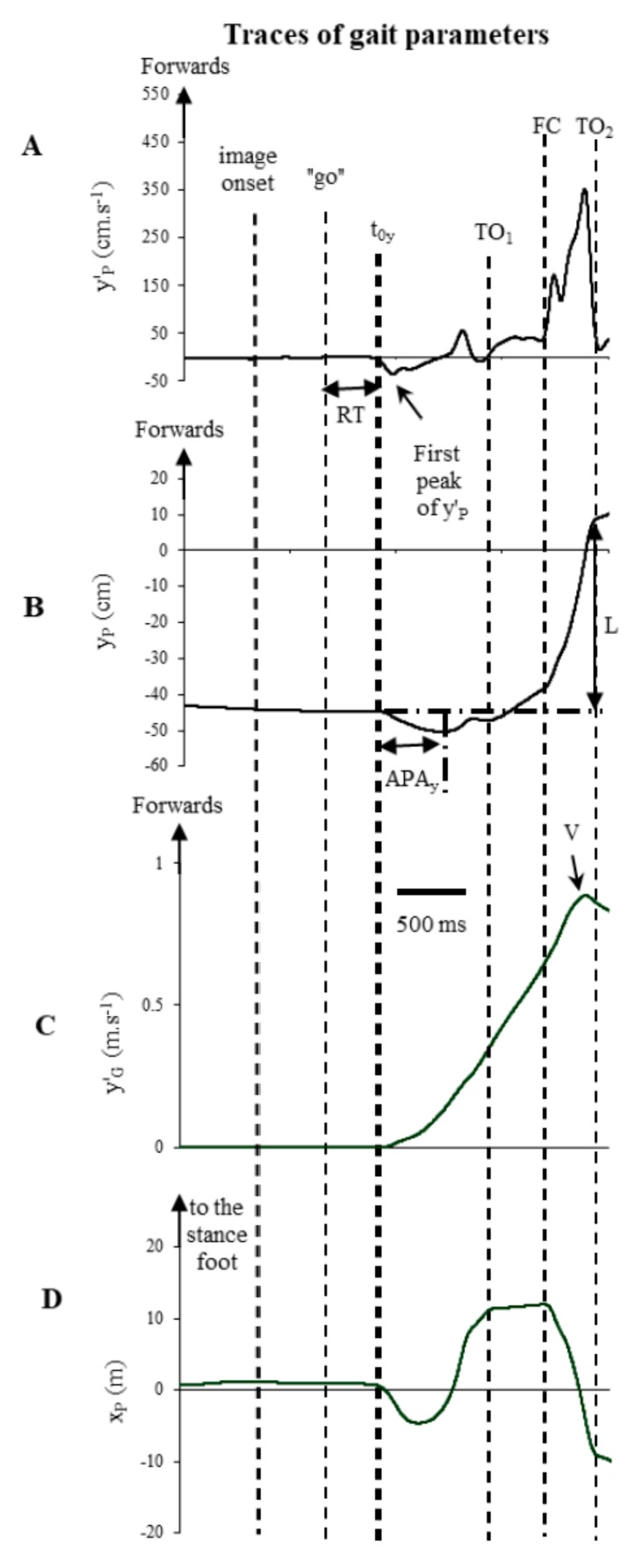
(**A**) anteroposterior velocity of CP (y’_p_); (**B**) anteroposterior displacement of CP (y_p_); (**C**) anteroposterior velocity of CM (y′_G_); (**D**) mediolateral displacement of CP (x_P_); one trail, one subject, during exposure of a pleasant image of the short condition. From top to bottom, the horizontal arrows correspond to the duration of anticipatory postural adjustments on the anteroposterior (APA_y_) and the vertical arrow to the first step length (L). TO_1_ is the time of foot contact of the stepping limb and TO_2_ is the time of toe off of the stance limb.

**Table 1 brainsci-08-00195-t001:** Mean and standard deviation of the parameters measured in both conditions with pleasant and unpleasant images.

Movement Parameters				
Condition	Short		Long	
Valence	Pleasant	Unpleasant	Pleasant	Unpleasant
t_yPmin_ (ms)	379 ± 69	360 ± 60	365 ± 63	360 ± 65
y_Pmin_ (cm)	−6.29 ± 2.07	−5.88 ± 2.25	−6.28 ± 2.23	−5.84 ± 2.20
V (m·s^−1^)	1.086 ± 0.133	1.058 ± 0.132	1.075 ± 0.121	1.054 ± 0.143
t_V_ (s)	1.102 ± 0.114	1.103 ± 0.140	1.113 ± 0.128	1.119 ± 0.146
L (cm)	59.69 ± 7.71	58.91 ± 7.14	59.02 ± 7.38	58.49 ± 8.07
MT (s)	3.061 ± 0.736	3.119 ± 0.751	3.090 ± 0.732	3.154 ± 0.768

From top to below, t_yPmin_ is the APAs duration along the antero posterior axis, y_Pmin_ the amplitude of the maximal backward shift of CP, V the peak of CM forward velocity, t_V_ the time for reaching the peak of forward CM velocity, L the length of the first step and MT the movement time and MT the movement time.

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
