# Peer review of "The Duration of Emotional Image Exposure Does Not Impact Anticipatory Postural Adjustments during Gait Initiation"

_brainsci, 2018, doi:10.3390/brainsci8110195_

Reviewer 1 Report

This is a very nicely done study. The kinematic and kinetic parameters reported in this study for gait initiation representative of APAs offer a sound evidence for the motivational direction hypothesis. I just have a few comments and minor edits, that are listed below. Nice work. 

Did the participants have prior exposure to such unpleasant or pleasant images? What if some participants were accustomed to some of these images?

Was there a self-reported grade for how the participants in this particular study perceived these images? In other words, did all participants perceive these pleasant and unpleasant images at the same intensity? How would the authors speculate this?

Were there any trials that were not successful?

Please make sure in Figure 1, even the TO1, FC and TO2 are explained in the figure description. These have been described in the text, but its ideal to be described again under the figure. 

I am not familiar with the end of two paragraphs in the introduction use the sentence, " we call it, ........" in describing the effects. Is this referring to hypothesizing? If yes, i would suggest the authors to write it this way in hypothesizing the effects. 

I mentioned earlier about prior exposure to similar pleasant or unpleasant images. Do participants tend to adapt to unpleasant images when shown in series one after another?

Overall, a nicely done and written study on an interesting topic.

Author Response

Did the participants have prior exposure to such unpleasant or pleasant images? What if some participants were accustomed to some of these images?

All participants were students of the University of Nanterre. None of them said that she (he) viewed one of the images we used in the study.

Was there a self-reported grade for how the participants in this particular study perceived these images? In other words, did all participants perceive these pleasant and unpleasant images at the same intensity? How would the authors speculate this?

As in our previous study [13], subjective ratings of valence and arousal of emotional images were obtained with the SAM procedure and the results were similar to our previous study [13]. We forgot to explain the role of the mouse, when the participant clicked on it at the end of the movement. So we added a sentence to explain that it triggered the SAM procedure (line 118 to121) - The results of the SAM procedure were added in the new manuscript (at the beginning of the results section) . Although the results of the subjective ratings of valence and arousal of emotional images was out the main aim of this study (APAs), we reasoned that it was better to add two sentences in the conclusion (line 307 to 311) to explain that the change in the subjective ratings of valence of unpleasant images (only in the long condition) might result from the implementation of an emotional regulation in this condition.

Were there any trials that were not successful?

From line 147 to 150, we wrote that 35 trials were rejected, but only for technical reasons.

Please make sure in Figure 1, even the TO1, FC and TO2 are explained in the figure description. These have been described in the text, but its ideal to be described again under the figure

This is done

I am not familiar with the end of two paragraphs in the introduction use the sentence, " we call it, ........" in describing the effects. Is this referring to hypothesizing? If yes, i would suggest the authors to write it this way in hypothesizing the effects.

We wrote it line 75 and 76.

I mentioned earlier about prior exposure to similar pleasant or unpleasant images. Do participants tend to adapt to unpleasant images when shown in series one after another?
Images were randomly presented both for valence, arousal and condition. Hence, we do not know whether an adaptation occurs when unpleasant images are presented in series one after another (in block).

Reviewer 2 Report

Review of: duration of emotional image exposure ......, by Gelat et al

This is a well designed study that closely follows existing paradigms, and with an impressive sample size. I am overall positive to the study, but I did find a substantial number of errors, inconsistencies, and odd wordings.

My comments are thus:

Line 35, what is a ‘mechanical step of the movement’? I guess the authors mean the first ‘segment’, or ‘phase’. Not ‘step’.

Line 46. Please add the word ‘system’ directly after ‘appetitive and defensive’, followed by a comma.

Line 73. Change ‘image’s  valence’ to ‘valence of the image’

Line 76/77 ‘We will call a condition effect the result that APAs on the progression axis would be larger when the body leaned forwards.’ This sentence is very odd, for numerous reasons. But what is more, I would not use the term ‘condition’ here or in the rest of the paper, because ‘condition’ usually refers to a generic factor, such as statistical conditions. Please be more specific, and label this for example ‘postural assymetry effect’, or something.

Line 83: These predictions are not really clear from the preceding text. Please make more specific, why this is the prediction.

Line 86. We read ‘because the body leaned’. But this is confusing because we are still dealing with a prediction. Perhaps ‘because the body was expected to lean’?

Line 91. The reference to study [16] is odd, because that study made use of manual responses, and not APAs

Methods: I presume the durations (long short) were randomized across trials, but this should be stated explicitly in the text.

Line 126. Please state that the normative picture values were adopted from the IAPS manual (I presume)

Line 134. Drop the word ‘finally’

Line 141 ‘calculated from the ground reaction forces and the movements measured by the force platform’. Again strange wording, because it suggests that forces and movements are two separate things.

Line 147. Consider deleting the phrase where RT is defined and move to the next Section (analysis)

Figure 1. Please consider adding labels (A, B, C and D) to the panels, and refer to these panels in the text. This will greatly help in clarification.

Figure 1: Please use somewhat larger font size, especially the vertical axis with all the subscripts and superscripts.

Line 184. As stated earlier, I find it confusing to speak of an effect of Condition; this is too generic.

Line 196. I am confused; shouldn’t CM be CP here?

Line 238. What is meant by ‘take advantage’? Is there a competition of some sort here?

Page 8, Discussion: please never use wordings like ‘we thought’. This is too informal. Use ‘we reasoned’, or ‘we assumed’ instead.

I found the final sentence (line 285 / 287) difficult to follow. Please consider a simple and more to-the-point wording of the conclusion.

Reference 13 (line 313), should read ‘reaction time’, not ‘time reaction’

Author Response

Importantly, because of the comments of the reviewer 1 we added a paragraph at the beginning of the results section entitled SAM (Self Assessment Manikin). Indeed, it was necessary to present the results of the SAM of this study (lines 183-203). As a result, two sentences were added (lines 307-311) in the conclusion to suggest that the change in the subjective ratings of valence of unpleasant images (only in the long condition) likely resulted from the implementation of an emotional regulation in this condition.

Line 35, what is a ‘mechanical step of the movement’? I guess the authors mean the first ‘segment’, or ‘phase’. Not ‘step’.

We only mean the movement onset; we wrote that "the anticiparory phase begins with a postural movement"

Line 46. Please add the word ‘system’ directly after ‘appetitive and defensive’, followed by a comma

This is done

Line 73. Change ‘image’s  valence’ to ‘valence of the image’

This is done

Line 76/77 ‘We will call a condition effect the result that APAs on the progression axis would be larger when the body leaned forwards.’ This sentence is very odd, for numerous reasons. But what is more, I would not use the term ‘condition’ here or in the rest of the paper, because ‘condition’ usually refers to a generic factor, such as statistical conditions. Please be more specific, and label this for example ‘postural assymetry effect’, or something.

This is done in lines 74 to 76

Line 83: These predictions are not really clear from the preceding text. Please make more specific, why this is the prediction.

We removed "we predicted" and we wrote "Moreover, RT would"...

Line 86. We read ‘because the body leaned’. But this is confusing because we are still dealing with a prediction. Perhaps ‘because the body was expected to lean’?

We now wrote: "because a previous study demonstrated that the body leaned forwards prior to the movement in the long condition"

Line 91. The reference to study [16] is odd, because that study made use of manual responses, and not APAs

The right reference is McKinnon et al. 2007 [16] (during gait initiation), we added in the references. the sentence line 90-91 has been changed. Neverthless, the reference of Carlsen [19] was kept to write (in the discussion line 280 to 282) about the high uncertainty  of the timing of the imperative go.

Methods: I presume the durations (long short) were randomized across trials, but this should be stated explicitly in the text.

This is written line  137-138

Line 126. Please state that the normative picture values were adopted from the IAPS manual (I presume)

This is already written line 124

Line 136. Drop the word ‘finally’

This is done

Line 144 ‘calculated from the ground reaction forces and the movements measured by the force platform’. Again strange wording, because it suggests that forces and movements are two separate things.

It's an error. We wanted to write moments and not movements measured by the plateform.

Line 147. Consider deleting the phrase where RT is defined and move to the next Section (analysis)

This is done

Figure 1: Please use somewhat larger font size, especially the vertical axis with all the subscripts and superscripts

We decided to not change the font size because a larger one would overlap the dashed vertical lines.

Line 184. As stated earlier, I find it confusing to speak of an effect of Condition; this is too generic.

We decided to not change the term condition because it was clearly defined in the introduction section.

Line 196. I am confused; shouldn’t CM be CP here?

At lines 164-165, we wrote that the CP position during a posture corresponds to the vertical projection of the CG.

Line 238. What is meant by ‘take advantage’? Is there a competition of some sort here?

Indeed, we wrote (lines 310-312): "The fact that the reaction time was similar with pleasant and unpleasant images in the long condition may be viewed as an advantage of this condition as compared to the short condition". The term of advantage was used in the sense that during the long condition participants were as fast with unpleasant than pleasant images during an approach behavior.

Page 8, Discussion: please never use wordings like ‘we thought’. This is too informal. Use ‘we reasoned’, or ‘we assumed’ instead

The correction is made

I found the final sentence (line 285 / 287) difficult to follow. Please consider a simple and more to-the-point wording of the conclusion.

The end of the conclusion means that although RT with pleasant and unpleasant images was found to be similar in the long condition, movement velocity was always faster with pleasant images, as compared to unpleasant images.

Reference 13 (line 313), should read ‘reaction time’, not ‘time reaction’

The correction is made
